# Self-Reported Practices in Opioid Management of Chronic Noncancer Pain: An Updated Survey of Canadian Family Physicians

**DOI:** 10.3390/jcm9103304

**Published:** 2020-10-14

**Authors:** Andrea D. Furlan, Santana Diaz, Angela Carol, Peter MacDougall, Michael Allen

**Affiliations:** 1KITE Toronto Rehabilitation Institute, University Health Network, Toronto, ON M5G 2A2, Canada; 2Institute for Work & Health, 400 University Ave Suite 1800, Toronto, ON M5G 1S5, Canada; 3Institute of Medical Sciences, University of Toronto, Toronto, ON M5S 1A8, Canada; santhanna.diaz@gmail.com; 4Family Medicine, Faculty of Health Sciences, McMaster University, Hamilton, ON L8P 1H6, Canada; acarol@cpso.on.ca; 5Department of Anesthesia, Pain Management & Perioperative Medicine, Dalhousie University, Halifax, NS B3H 4R2, Canada; pcmacdou@gmail.com; 6Continuing Professional Development, Dalhousie University, Halifax, NS B3H 4R2, Canada; michael.allen@dal.ca

**Keywords:** opioids, guidelines, family physicians, primary care, survey

## Abstract

Chronic pain affects one in five Canadians, and opioids continue to be prescribed to 12.3% of the Canadian population. A survey of family physicians was conducted in 2010 as a baseline prior to the release of the Canadian Opioid Guideline. We repeated the same survey with minor modifications to reflect the updated 2017 opioid prescribing guideline. The online survey was distributed in all provinces and territories in both English and French. There were 265 responses from May 2018 to October 2019, 55% of respondents were male, 16% had advanced training in pain management, 51% had more than 20 years in practice, 54% wrote five or fewer prescriptions of opioids per month, and 58% were confident in their skills in prescribing opioids. Of the 11 knowledge questions, only two were correctly selected by more than 80% of the respondents. Twenty-nine physicians (11%) do not prescribe opioids, and the main factor affecting their decisions were concerns about long-term adverse effects and lack of evidence for effectiveness of opioids in chronic noncancer pain. Of the 12 guideline-concordant practices, only two were performed regularly by 90% or more of the respondents: explain potential harms of long-term opioid therapy and beginning dose of less than 50 mg of morphine equivalent daily. This survey represents a small proportion of family physicians in Canada and its generalizability is limited. However, we identified a number of opioid-related and guideline-specific gaps, as well as barriers and enablers to prescribing opioids and adhering to the guideline.

## 1. Introduction

Chronic noncancer pain (CNCP) continues to be a major health problem affecting one in five Canadians [1]. Opioids are frequently prescribed to alleviate pain and improve function in patients with CNCP, although there is limited evidence for effectiveness beyond three months of long-term opioid therapy [2].

In Canada, from 2013 to 2018 there were fewer people being prescribed opioids (14.3% to 12.3%), fewer people starting opioids (9.5% to 8.1%), fewer people being prescribed opioids on a long-term basis (19.8% to 17.6%), more people stopping long-term opioid therapy (18.3% to 20.4%), and more people on long-term opioid therapy were being prescribed smaller doses (72.1% to 76.3%). However, the dosage and duration of opioid therapy among people starting opioids remained relatively stable [3].

Despite the reduction of available opioids in Canada, the number of opioid-related emergency visits, overdoses, and deaths continues to climb. Between January 2016 and December 2019 there were 15,393 apparent opioid-related deaths in Canada, with recent estimates indicating that 9.6% of Canadian adults who use opioid medications in 2018 reported some form of problematic use [4,5]. However, among people who died from illicit drug overdose in the Canadian Province of British Columbia between 2015 and 2017, 85.5% had opioids relevant to death on toxicology; of these, both prescribed-only opioids (2.4%) and a combination of prescribed and nonprescribed opioids (7.8%) were relatively rare, suggesting that the majority of opioids are from illicit sources [6].

A survey of Canadian family physicians practices in opioid management of CNCP was conducted in 2010 [7], the year the first Canadian Opioid Guideline was released by the National Opioid Use Guideline Group (NOUGG) [8]. In 2017, an update for the guideline was released [9]. 

The objectives of this study are to repeat the survey to (1) determine current physicians’ practices and knowledge in prescribing opioids for CNCP in relation to the updated Canadian opioid guideline; (2) identify changes from the survey conducted in 2010; and (3) determine adherence to the guideline, and barriers and facilitators for physicians in prescribing opioids for CNCP and adhering to the guideline’s recommendations. 

## 2. Methods and Materials

### 2.1. Survey Methods

This survey included family physicians who manage patients with CNCP and who practice medicine in any Canadian province. An online survey in English and French was hosted at Survey Monkey^®^ and was sent to the medical regulatory authorities of Canada and the College of Family Physicians of Canada for distribution. These organizations used electronic and print newsletters, magazines, and social media with embedded links to the survey to their respective constituents. The number of physicians who received the invitation to complete the survey is unknown, but we know that in 2018, there were approximately 45,000 family medicine physicians in Canada [10]. Given the lack of a discrete sampling frame and the varied methods of contacting family physicians, a nonprobability convenience sample was obtained. 

The invitation and introduction to the survey specified that participants consent to their participation by answering the questions provided and gave the option to exclude themselves if they do not prescribe opioids for CNCP. In this case, they would skip most questions about self-reported practices and would only answer questions about knowledge, barriers, and facilitators. Examples of weak and strong opioids were provided: weak opioids—codeine, tramadol, propoxyphene, meperidine. and pentazocine; strong opioids—morphine, oxycodone, hydromorphone, transdermal fentanyl, and methadone. 

The survey was open from May 2018 to October 2019. There was no incentive for completing the survey. To identify rural or urban settings, a question asked if the second digit of their postal code was zero (rural) or not zero (urban). The University of Toronto Research Ethics Board approved the study. 

### 2.2. Modifications to the 2010 Survey

We repeated the 2010 online survey with minor revisions to some questions, and we added questions that are related to new topics included in the 2017 guideline. The methods and results of the 2010 survey are described elsewhere [7]. 

The main changes to the survey were related to the watchful dose and opioid tapering. The 2010 guideline had introduced the term “watchful dose” of opioids, the daily dose of 200 mg morphine equivalents at which patients may need to be reassessed or more closely monitored. The 2017 guideline eliminated the term and changed the dose to 90 mg of morphine equivalents per day or more. The 90 mg dose was introduced as a “strong recommendation” for CNCP patients beginning long-term opioid therapy to restrict the dose to less than 90 mg of morphine equivalents daily rather than having no upper limit or a higher limit on dosing. In addition, there was another “weak recommendation” for patients with CNCP who are currently using 90 mg morphine equivalents daily or more to taper the opioid to the lowest effective dose, potentially discontinuing the opioid therapy. 

In the demographic characteristics section, we added one new question: “Wait time for second opinion regarding 90 mg morphine equivalent daily”, and we added an alternative response in three questions: “I don’t have this available” (Table 1).

There were three new questions added to the knowledge section: One new open-ended question about the minimum daily dose of opioid in morphine equivalent that the patient would be taking before the physician would prescribe transdermal fentanyl. This topic was related to the 2017 opioid guideline. The other two new questions were not directly related to the opioid guideline (opioid replacement therapy and one about medical cannabis). These two questions were included as the members of our team who work with regulatory authorities are collaborating in the development of educational materials for physicians. 

We had noticed that in the 2010 survey 65% of respondents selected the wrong answer to what was considered a clinically significant reduction in pain intensity, perhaps because this was a tricky question, as the opioid guideline recommended a 30% reduction in pain be considered clinically significant. We therefore revised this knowledge question “A 20% reduction in pain intensity is considered clinically significant” in the 2010 survey to “A 30% reduction in pain intensity is considered clinically significant” in the 2018 survey. 

### 2.3. Data management and Analyses

Questions regarding physicians’ practices listed recommended practices and asked respondents how frequently they performed each practice (never, <25%, <50%, >50%, >75%, and always). For these questions, the percentage of respondents performing these practices is reported in three categories: never and <25% of patients, 25% to 75% of patients, and >75% and always. 

Questions regarding physicians’ knowledge asked if they agreed, disagreed, or had no opinion about various statements. 

Questions regarding barriers and enablers to prescribing opioids and adhering to the guideline asked respondents to rate the importance of various factors on a 5-point Likert scale (1 = not very important, 5 = very important). For each factor, the per cent of response is reported in three categories: 1 and 2 (not important), 3 (neutral), and 4 and 5 (important). 

Analysis was performed with Microsoft Excel^®^. We considered a difference of absolute 10 percentage points or larger to be meaningful between the surveys in 2010 and 2018. The complete survey is available in English and French (Appendix A). 

## 3. Results

### 3.1. Responses

We received 265 responses, all of which were in English. Responses according to province were Saskatchewan, *n* = 117 (63%); Alberta, *n* = 30 (16%); British Columbia, 26 (14%); Ontario, 8 (4%); Manitoba, 2 (1%); Nova Scotia, 1 (1%); and Yukon, 1 (1%). There were no responses from New Brunswick, Newfoundland and Labrador, Northwest Territories, Nunavut, Prince Edward Island, or Quebec. Fifty-six of the 185 (30%) respondents were from rural areas. 

Demographic and practice variables for all participants are shown in Table 1. Not all respondents answered all questions. Fifty-five per cent were male, 51% had more than 20 years in practice, 65% see less than 400 patients per month, 54% wrote five or fewer prescriptions for opioids per month, and 58% were confident in their skills in prescribing opioids for CNCP. 

### 3.2. Access to Physicians with Expertise in Pain and/or Addiction Medicine

Twenty-four percent responded that the wait time for nonurgent referral to a pain specialist is longer than 12 months and 23% do not have access to a pain specialist. Regarding access to a second opinion about patients on high dose opioids (>90 mg of morphine equivalent daily), 23% of respondents do not have this available to them, and 22% responded they would have this available in less than one month. The wait time for nonurgent referral to addiction medicine was less than 6 months among 44% of respondents; however, 15% do not have access to this service (Table 1).

### 3.3. Knowledge of Opioids

Table 2 shows responses to the 11 knowledge questions. Only two questions were correctly selected by more than 80% of the respondents. The proportion of correct answers varied from 23% to 89% with an average of 55%. 

The question about the minimum daily dose of opioid in morphine equivalent that the patient would be taking before the physician would prescribe transdermal fentanyl was correctly answered by 80 physicians (48%). Thirty-five physicians (21%) incorrectly responded that “there is no minimum dose, it varies with patient condition”, 12 physicians (7%) incorrectly responded that the minimum is 40 mg, and 7 physicians (4%) incorrectly responded that the minimum was 20 mg. Twenty percent of physicians (*n* = 34) had no opinion about this question. No physician responded that transdermal fentanyl is their first line of opioid for CNCP. 

There was one new question in the 2018 survey that showed a high response of physicians with “no opinion”, which was 39% to “Medical cannabis is effective for neuropathic pain”. Only 35% selected the correct response of “agree” with this statement.

### 3.4. Opioid Prescribing Practices

Seventy percent (*n* = 185) of respondents prescribed both weak and strong opioids. Eleven percent (*n* = 29) did not prescribe opioids, 16% (*n* = 43) prescribed only weak opioids, and 3% (*n* = 8) prescribed only strong opioids. 

Table 3 shows the results regarding factors affecting decisions not to prescribe opioids for CNCP among the 29 physicians who answered: “I do not prescribe opioids for CNCP.” The most important factors affecting their decisions were concerns about long-term adverse effects (96%) and lack of evidence for effectiveness of opioids in CNCP (79%). Importantly, concerns about becoming a “target prescriber” or audit from regulatory or monitoring bodies were not major barriers, respectively, rated by 32% and 18% of the respondents.

Results regarding factors affecting decisions not to prescribe strong opioids for CNCP among the 43 physicians who answered: “I only prescribe weak opioids for CNCP” are in Appendix A. The most important factors affecting their decisions were concerns about long-term adverse effects, belief that strong opioids are commonly diverted and abused in community, and lack of evidence for effectiveness of strong opioids in CNCP. It is interesting that concerns about becoming a “target prescriber” or audit from regulatory or monitoring bodies were rated by 65% and 51% by the respondents, respectively. 

### 3.5. Adherence to Opioid Prescribing Guideline

The 2017 Canadian opioid guideline defines CNCP as “any painful condition that persists for ≥3 months that is not associated with a diagnosis of cancer”. Only 17% (*n* = 46) of the respondents responded that this definition is similar to their own definition of CNCP. The majority of responses (*n* = 159, 60%) said their definition of chronic pain is “pain persisting beyond the time normally associated with healing for a specific illness or injury”, and 60 (23%) responded that they use “pain that persists more than 6 months”. 

The 29 physicians who do not prescribe opioids for CNCP were automatically skipped from seeing the following sections leaving an eligible sample of 236 participants.

Table 4 shows the frequency of following 12 guideline-concordant practices performed before starting patients on opioids for CNCP. Fourteen practices are shown, two are new in relation to the survey conducted in 2010, and two are distracters (practices not recommended in the guideline) to reveal whether respondents tended to report they always performed the listed practices. Only two practices were performed regularly by 90% or more of the respondents: explain potential harms of long-term opioid therapy and beginning dose of less than 50 mg of morphine equivalents daily. The number of participants who perform these practices regularly ranged from 27% to 95%, with an average of 66%. The two distracter questions were not performed by 122 (59%) and 116 (56%) of the respondents, suggesting the participants were paying attention to the survey questions. 

Table 5 shows the frequency of following 13 guideline-concordant practices performed while monitoring patients on opioids for CNCP. Only one practice was performed regularly by 90% of more of the participants: observing for aberrant drug-related behavior. Three practices were performed regularly by 80% to 90% of the participants: assessment for specific adverse effects, advising the patient to use caution while driving or operating machinery, and assessing the patient’s level of function. The new practice added to this survey was performed regularly by 70 (42%) respondents, and it is related to referring the patient to formal multidisciplinary program if the patient is experiencing a serious challenge in tapering opioid. 

The daily dose of morphine equivalent that the respondents considered that the patient might need to be referred for a second opinion was 90 mg by 44 physicians (25%), 100 mg by 31 physicians (20%), and 200 mg by 12 physicians (15%). The average was 93 mg and the median 90 mg (See Appendix A)

### 3.6. Enabling Factors for Prescribing Opioids and for Adherence to Opioid Prescribing Guideline

Table 6 shows physicians’ ratings of factors for optimizing use of opioids for CNCP. The top 3 highest-rated factors were access to patient’s opioid prescription history from a provincial monitoring program (rated useful by 88% of physicians), followed by improved access to consultants who are experts in pain or addiction (87%) and availability of non-pharmacological options for treating CNCP (85%). The latter was one of the new items added to this updated survey. The other new item added to this survey was rated somewhat lower (72%): accessibility of other pharmacological agents (transdermal or sublingual buprenorphine). 

## 4. Discussion

This study is an updated survey of family physicians’ practices in opioid management of CNCP in Canada. The demographic characteristics of the physicians who responded to this survey were similar to the physicians who answered the 2010 survey, except for provincial distribution. In 2010, 52% were from Ontario, and in 2018, 63% were from Saskatchewan. Given the non-probabilistic nature of the sample, we suggest caution in generalizing to the larger population of Canadian family physicians, as the 2018 survey might be susceptible to low response bias. The respondents of this survey were mostly male, with more than 20 years in practice, working in urban settings with small size practices. Most respondents write few prescriptions of opioids per month and they are very confident in their skills to prescribe opioids. The respondents have difficulty accessing services with expertise in pain and addiction medicine. 

The average of correct responses in knowledge regarding opioids for CNCP was 55% (range 23% to 89%), which has improved from 2010 where the average of participants who had achieved correct responses was 38% (range from 13% to 81%) [7]. Another important aspect of knowledge is to safely switch from an oral opioid to transdermal fentanyl. In 2010, 38% of the respondents correctly identified the minimum daily dose of opioid a patient should be taking before receiving the transdermal fentanyl (60 mg). In the present survey, this number increased to 48% of the respondents. Similarly, in the current survey 65% of respondents correctly disagreed with the statement that patients could be safely switched from a high dose of codeine to transdermal fentanyl compared with 39% in 2010. While these results may indicate better knowledge translation, exchange, and education regarding safe practices, they are still problematic, considering the seriousness of the situation. The new question about medical cannabis being effective for neuropathic pain showed a low rate of 35% of correct responses, and this will be important when we repeat this survey in the future to assess if this rate is improved in Canada.

Other improvements in knowledge regarding opioid use from 2010 are correctly disagreeing that some strong opioids provide greater pain relief than others (42% correctly answered in 2018/19 vs. 21% in 2010), controlled release opioids are more effective in controlling pain than immediate release opioids (38% correctly answered in 2018/19 vs. 27% in 2010), and there is RCT evidence that opioids are effective in long-term relief of CNCP (53% correctly disagreed in 2018/19 vs. 13% in 2010).

It is possible that improved knowledge about the lack of RCT evidence for long-term relief of CNCP leads to more physicians not prescribing opioids or prescribing only weak opioids. Twenty-seven percent of family physicians surveyed do not prescribe opioids or prescribe only weak opioids an increase from the 13% found in 2010. The unintended consequence of this finding is that it may become difficult for some patients who need legitimate opioid prescriptions to find a family physician to prescribe for them. A recent survey in Nova Scotia showed that 28% of family physicians accepting new patients would not accept a new patient if the person is on opioids [11]. 

In our study, the top barriers for not prescribing opioids were similar to the barriers reported in 2010 and included concerns about long-term adverse effects (95% in 2018 and 87% in 2010) and lack of evidence for effectiveness of opioids in CNCP (79% in 2018 and 66% in 2010). More physicians were concerned about short-term adverse effects (43% in 2018 vs. 19% in 2010), but fewer were concerned that patients complain of pain out of proportion to objective findings (32% in 2018 vs. 63% in 2010). 

The top barriers affecting their decisions to prescribe only weak opioids (e.g., codeine or tramadol) were also similar to 2010 and included concerns about long-term adverse effects (88% in 2018 and 88% in 2010), belief that strong opioids are commonly diverted and abused in community (75% in 2018 and 83% in 2010), and concern about becoming a target prescriber of opioids (65% in 2018 and 60% in 2010). More physicians were concerned about the lack of evidence for effectiveness of strong opioids in CNCP (73% in 2018 vs. 47% in 2010).

With regards to adherence to the Canadian opioid guideline, it was expected that in 2018 a great majority of Canadian family physicians would have been exposed for enough time to the 2010 guideline, and for at least one year to the updated 2017 guideline. It was surprising to find that the respondents’ definitions of chronic non-cancer pain matched the guideline by only 17% of the respondents. 

With respect to practices performed regularly before starting patients on opioids, the results showed that the top guideline-concordant practice in both surveys was to explain potential harms of long-term opioid therapy (95% in 2018 and 87% in 2010). However, the second and third top practices in 2018 were beginning opioid therapy with less than 50 mg morphine equivalent per day (91%, new practice included only in the 2018 survey), and assessing for past/current substance use disorder as well as an active psychiatric disorder (86%, new practice included only in the 2018 survey). 

Several recommended practices before starting patients on opioids from the 2010 guideline were performed more frequently in the current survey: use of a signed treatment agreement increased from 37% to 62%; assessing patient’s level of pain using a scale increased from 47% to 60%; tapering patients off benzodiazepines increased from 44% to 59%, assessing risk of addiction using a screening tool increased from 37% to 51%; and perhaps most notably urine drug screening increased from 15% to 44%. These increases suggest that it takes many years for guidelines’ recommendations or guidance statements to be implemented in practice.

With respect to practices performed while monitoring patients on opioids, the top three guideline-concordant practices were exactly the same as in 2010: observe for aberrant drug-related behavior (95% in 2018 and 93% in 2010), assess for adverse effects (89% in 2018 and 84% in 2010), and advise caution while driving or operating machinery (88% in 2018 and 82% in 2010). Again, urine drug screening increased markedly in 2018, from 22% in 2010 to 57% in the current survey. However, if the patient was having insufficient pain relief physicians were less likely to increase the dose of opioid (28% in 2018 vs. 53% in 2010) or try a different opioid (23% in 2018 vs. 40% in 2010). These changes and lack of changes are reassuring that physicians are following the 2017 guideline’s recommendations that are much more restrictive than the 2010 guideline. 

Regarding knowledge of what is the daily dose of opioids they consider a need for a second opinion, only one in four physicians agreed with the 2017 guideline that it should be 90 mg of morphine equivalents daily. The remaining physicians thought it should be higher than 90. 

Several recommended practices were reported as having been performed more frequently since the first survey, most notably urine drug screening and use of management agreements. However, these practices are not classified as recommendations in the 2017 guideline. Rather, they are the subject of “guidance statements” that describe the uncertainty around their effectiveness. 

There are many reasons why physicians may not adhere to opioid guidelines. A recent qualitative study conducted with family physicians in Ontario looked at facilitators and barriers in using the Canadian opioid guideline, and found that the guidelines were practical and pragmatic and constitute a “safety net” or “framework”, while the main barriers were time and effort required to become familiar with the contents and “very labor-intensive” [12]. It is also noteworthy that much of the evidence around opioid prescribing is of low to moderate quality. The current guideline makes 10 recommendations of which only four are “strong recommendations” so physicians may feel less compelled to adhere to guidelines. Strong recommendations indicate that all or almost all fully informed patients would choose the recommended course of action which would indicate to clinicians that the recommendation is appropriate for all or almost all individuals [9].

Our survey also identified potential enablers to effective opioid prescribing for CNCP. Similar to the survey conducted in 2010, the top two factors were access to patients’ opioid prescription history from provincial monitoring program (88% in 2018 and 87% in 2010), and improved access to pain or addiction specialists (87% in 2018 and 84% in 2010). The third top factor in 2018 was availability of non-pharmacological options (85%, a new factor added only in the 2018 survey). The other factor that tied in second in 2010 was knowledge of risks and benefits of different opioids (84%).

The main limitations of this study are very similar to the limitations reported in 2010. All response are self-reported practices, however, respondents reported infrequently conducting practices included as “distracters”, suggesting credence to the findings. The number of responses represents a small proportion of the 45,000 family physicians who practice in Canada. We received no response of the French version of the survey. The geographical distribution was unbalanced with majority of respondents from the province of Saskatchewan. We used a non-probability convenience sample which limits the generalizability of our findings. In the absence of a clear definition of what is a pain specialist or an addiction specialist, it is difficult to validate the answers about wait times to specialists. Finally, we did not ask questions about respondents’ exposure to or use of the guideline or their opinions of it.

The demographics of the physicians who participated in this survey have similarities and differences with the 2018 demographics distribution of Canadian family physicians: In our survey, 55% were male, compared to 53.4% in Canada [10]. However, in our survey, 30% work in rural settings, compared to only 8% in Canada [13]. 

This survey provides a snapshot of family physicians’ current opioid-prescribing practices, opioid guideline knowledge and concordance. One very serious knowledge gap is the lack of understanding about switching patients to transdermal fentanyl, a very potent, high dose opioid that has the potential to cause overdose quickly. It is very concerning that there are still 52% of respondents who did not get the correct response to this question. This indicates there is still a great need for knowledge translation, exchange, and education in how to prescribe opioids for patients with chronic pain. 

It would be informative to repeat this survey at regular intervals in two to five years to detect changes over time; however, this will have to take account of the changes in format and recommendations of updated guidelines. These national surveys are attempts to assess if the release of opioid guidelines has been achieving the changes that they proposed to make. If the survey is repeated regularly it may be possible to establish ongoing relationships with provincial and national organizations to distribute it more widely and encourage participation. It is important to seek support and endorsement of organizations such as the College of Family Physicians of Canada and the medical regulatory authorities. 

## 5. Conclusions

This survey represents a small proportion of family physicians in Canada and its generalizability is limited. However, we identified a number of opioid-related and guideline-specific knowledge gaps, areas where the Canadian opioid guideline’s recommendations are not being adhered to, barriers and enablers to prescribe opioids, and to adherence to the guideline. 

## Figures and Tables

**Table 1 jcm-09-03304-t001:** Demographics and practice characteristics of respondents.

Response	2018		2010	
	*n* (%)	Total Responses, *n*	%	Total Responses, *n*
**Sex**		185		622
Male	103 (55)		59	
Female	81 (44)		41	
Prefer not to answer	1 (1)		NA	
Have advanced training in pain management		185		627
	29 (16)		15	
**Years in practice**		185		621
1–5	31 (17)		17	
6–10	18 (10)		9	
11–20	41 (22)		18	
21–30	49 (26)		31	
>30	46 (25)		26	
**Population of practice community**		185		622
<5000	30 (16)		13	
5000–25,000	43 (23)		22	
>25,000–100,000	23 (12)		14	
>100,000–500,000	62 (34)		26	
>500,000	27 (15) ^†^		26	
**Patients seen per month**		185		592
<200	53 (29)		23	
200–400	67 (36)		33	
>400–600	41 (22)		29	
>600–800	18 (10)		9	
>800	6 (3)		6	
**Prescriptions for weak opioid written per month**		168		578
1–5	90 (54)		31	
6–10	43 (26)		31	
11–20	19 (11)		22	
>20	16 (10)		16	
**Prescriptions for strong opioid written per month**		166		548
1–5	90 (54)		46	
6–10	46 (28)		28	
11–20	16 (10)		14	
>20	14 (8)		12	
**Confidence prescribing opioids for chronic noncancer pain**		265		704
1 Not very confident	8 (3)		3	
2	28 (11)		8	
3	76 (29)		31	
4	111 (42)		43	
5 Very confident	42 (16)		15	
**Wait time for nonurgent referral to pain specialist, months**		185		609
<1	4 (2)		3	
1–6	40 (22)		23	
>6–12	45 (24)		28	
>12	44 (24)		39 ^†^	
Don’t know	10 (5)		7	
I don’t have this available	42 (23)		NA	
**Wait time for second opinion regarding 90 mg morphine equivalent daily**		185		NA
<1	40 (22)		NA	
1–6	32 (17)		NA	
>6–12	19 (10)		NA	
>12	14 (8)		NA	
Don’t know	38 (21)		NA	
I don’t have this available	42 (23)		NA	
**Wait time for nonurgent referral to addiction specialist, months**		185		623
<1	25 (14)		7	
1–6	55 (30)		26	
>6–12	18 (10)		21 ^†^	
>12	26 (14)		18	
Don’t know	34 (18)		28 ^†^	
I don’t have this available	27 (15)		NA	

NA: not applicable, this was a new question included only in the 2018 survey. ^†^ Difference is 10% or larger.

**Table 2 jcm-09-03304-t002:** Knowledge regarding opioid use in chronic noncancer pain (CNCP). Shown in decreasing order of correct answers in the 2018 survey.

	Year	Frequency of Response, %			Total Responses, *n*
		Disagree	Agree	No Opinion	
A 20% reduction in pain intensity is considered clinically significant (2010). A 30% reduction in pain intensity is considered clinically significant (2018).	2010 2018	18 * 5 ^†^	65 89 *^,†^	17 6 ^†^	604 167
Pain relief is a more important indicator of opioid effectiveness than functional ability	2010 2018	81 * 86 *	11 7	9 7	604 167
(NEW) Opioid replacement therapy is effective for patients with opioid abuse disorder	2010 2018	NA 14	NA 71 *	NA 15	NA 168
There is evidence from RCTs that opioids are effective in short-term (up to 3 months) relief of CNCP	2010 2018	8 17	75 * 68 *	17 15	603 168
Patients may be safely switched from a high dose of codeine to a fentanyl patch	2010 2018	39 * 65 *^,†^	46 16 ^†^	16 18	598 168
There is evidence from RCTs that opioids are effective in long-term (over 3 months) relief of CNCP	2010 2018	13 * 53 *^,†^	69 33 ^†^	17 14	603 167
Some strong opioids provide better pain relief that others	2010 2018	21 * 42 *^,†^	71 48 ^†^	9 11	603 168
Controlled-release opioids are more effective in controlling pain than immediate-release opioids	2010 2018	27 * 38 *^,†^	63 50 ^†^	10 11	602 168
Controlled-release opioids have a lower risk of addiction than immediate release opioids	2010 2018	30 * 35 *	64 58	6 8	605 168
(NEW) Medical cannabis is effective for neuropathic pain	2010 2018	NA 26	NA 35 *	NA 39	NA 168
Some strong opioids are more likely to lead to addiction than others	2010 2018	28 * 23 *	63 72	9 5	603 168

Correct answers (*). RCT: Randomized controlled trial. ^†^ Difference is 10% or larger. “NEW” indicate new statements added to survey in conducted in 2010. NA: not applicable.

**Table 3 jcm-09-03304-t003:** Rating of factors affecting decision not to prescribe opioids for chronic noncancer pain (CNCP). Shown in decreasing order of importance in the 2018 survey.

	Year	Rating *, %			Total Responses, *n*
		Not Important	Neutral	Important	
Concern about long-term adverse effects, eg, addiction or misuse	2010 2018	7 4	7 0	87 96	31 28
Lack of evidence for effectiveness of opioids in chronic noncancer pain	2010 2018	16 4 ^†^	16 11	66 79 ^†^	32 28
Concern about short-term adverse effects, e.g., constipation, sedation	2010 2018	47 28 ^†^	31 25	19 43 ^†^	32 28
Type of practice limits follow-up e.g., walk-in clinic	2010 2018	43 47	10 18	40 33	30 28
Concern that patients complain of pain out of proportion to objective findings	2010 2018	16 25	22 36 ^†^	63 32 ^†^	32 28
Concern about becoming a “target prescriber” of opioids	2010 2018	34 60 ^†^	22 7 ^†^	38 32	32 28
Concern about audit from regulatory or monitoring body	2010 2018	56 57	19 25	22 18	32 28
Takes too much time to titrate and monitor	2010 2018	66 75	16 7	16 15	32 28
Inadequate knowledge of which opioids to use	2010 2018	72 75	16 14	6 7	32 28
Inadequate knowledge of dosages	2010 2018	78 79	13 18	6 0	32 28

* Percent of respondents rating importance of factor as 1 or 2 (not important), 3 (neutral), or 4 or 5 (important) on 5-point Likert scale. ^†^ Difference is 10% or larger. Percentage may not total 100% because some respondents indicated “no opinion”. The total of eligible participants to answer this question is 29 (“I do not prescribe opioids for CNCP”).

**Table 4 jcm-09-03304-t004:** Frequency of following recommended practices performed before starting patients on opioids (shown in decreasing order of frequency in the 2018 survey).

	Year	Frequency of Responses, %			Total Responses, *n*
		Never, <25%	25% to 75%	>75%, Always	
Explain potential harms of long-term opioid therapy	2010 2018	2 1	11 4	87 95	661 208
(NEW) Be sure to prescribe a dose less than 50 mg morphine equivalents daily	2010 2018	NA 4	NA 5	NA 91	NA 207
(NEW) Assess for past/current substance use disorders as well as active psychiatric disorders.	2010 2018	NA 1	NA 13	NA 86	NA 208
Assess patient’s level of function (e.g., social, recreational, occupational)	2010 2018	4 2	20 19	76 78	671 208
Explain potential benefits of long-term opioid therapy	2010 2018	9 10	17 16	75 74	665 208
Confirm that the patient has a condition that has been shown to benefit from opioids	2010 2018	11 10	27 27	62 63	654 204
Have patient sign a treatment agreement	2010 2018	42 15 ^†^	21 23	37 62 ^†^	665 207
Assess patient’s level of pain intensity using a scale	2010 2018	27 15 ^†^	26 25	47 60 ^†^	667 208
If patient is on a benzodiazepine, try to taper them off	2010 2018	21 11 ^†^	35 30	44 59 ^†^	650 208
Assess risk of addiction using screening tool	2010 2018	38 24 ^†^	25 25	37 51 ^†^	666 207
Perform urine drug screening	2010 2018	68 31 ^†^	17 24	15 44 ^†^	667 207
Give the patient written information about opioid therapy	2010 2018	62 47 ^†^	23 26	16 27 ^†^	659 208
(#) Refer to colleague for assessment	2010 2018	57 59	32 32	11 9	655 207
(#) Conduct formal psychological screening	2010 2018	71 56	18 25	11 19	668 208

Percent of respondents indicating they perform practices never or in <25% of their patients, in 25% to 50% of their patients, or in >75% of their patients or always. ^†^ Difference is 10% or larger. Number of respondents per question varied from 204 to 208 physicians. (NEW) New questions added to the survey. # Practices not recommended in guideline are included in survey as distracters to reveal whether respondents tended to report they always performed the listed practices. NA: not applicable.

**Table 5 jcm-09-03304-t005:** Frequency of following recommended practices performed while monitoring patients on opioids (shown in decreasing order of frequency in the 2018 survey).

	Year	Frequency of Responses, %			Total Responses, *n*
		Never, <25%	25% to 75%	>75%, Always	
Observe for aberrant drug-related behavior such as requesting higher doses or accessing opioids from other sources	2010 2018	2 1	6 3	93 96	651 168
Assess for specific adverse effects (e.g., nausea, constipation, drowsiness, dizziness)	2010 2018	3 1	13 10	84 89	648 168
Advise the patient to use caution while driving or operating machinery.	2010 2018	5 1	14 11	82 88	647 168
Assess patient’s level of function (e.g., social, recreational, occupational)	2010 2018	4 2	19 15	77 83	652 168
If patient has unacceptable side effects, try a lower dose	2010 2018	14 8	34 30	53 61	645 168
If patient has unacceptable side effects, try a different opioid	2010 2018	7 8	30 33	63 60	649 168
Assess patient’s level of pain intensity using a scale	2010 2018	28 17 ^†^	25 26	47 57 ^†^	652 168
Do routine or urine drug screening	2010 2018	58 20 ^†^	20 24	22 57 ^†^	653 168
(NEW) If patient is experiencing a serious challenge in tapering, try a referral to a formal multidisciplinary program	2010 2018	NA 35	NA 23	NA 42	NA 167
If patient has insufficient pain relief, taper off opioid and try another modality	2010 2018	26 15 ^†^	47 50	27 35	643 168
If patient has insufficient pain relief, increase the dose	2010 2018	4 13	43 59 ^†^	53 28 ^†^	647 166
If patient has insufficient pain relief, try a different opioid	2010 2018	7 19 ^†^	30 58 ^†^	63 23 ^†^	649 166
Ask patient to bring remaining medication to check compliance with the prescription	2010 2018	44 49	29 32	28 19	646 167

Percent of respondents indicating they perform practices never or in <25% of their patients, in 25% to 50% of their patients, or in >75% of their patients or always. ^†^ Difference is 10% or larger. Number of respondents per question varied from 166 to 168 physicians. (NEW) New questions added to the survey. NA: not applicable.

**Table 6 jcm-09-03304-t006:** Usefulness of enabling factors for optimizing use of opioids for chronic noncancer pain (shown in decreasing order of usefulness in the 2018 survey).

		Not Useful	Neutral	Useful	Total Responses, *n*
Access to patients’ opioid prescription history from provincial monitoring program	2010 2018	5 2	4 1	87 88	646 168
Improved access to consultants who are experts in pain or addiction	2010 2018	5 6	8 3	84 87	646 168
(NEW) Availability of non- pharmacological options	2010 2018	NA 4	NA 5	NA 85	NA 167
Knowledge of risks and benefits of different opioids	2010 2018	4 5	10 11	84 82	650 168
Tips in recognizing patients at high risk of addiction	2010 2018	6 8	11 8	83 82	651 168
Up to date guideline on use of opioids in CNCP	2010 2018	5 5	11 9	82 82	646 167
Validated scale to assess function (e.g., social, recreational, functional)	2010 2018	8 8	9 4	81 82	650 168
Continuing medical education in optimal use of opioids in CNCP	2010 2018	7 4	13 9	79 82	643 166
Knowledge of practical aspects of urine drug screening (e.g., collection sample, interpreting results)	2010 2018	13 7	11 7	72 82 ^†^	649 168
Availability of urine drug screening at local lab	2010 2018	18 9	15 8	64 81 ^†^	650 168
Patient education material	2010 2018	7 5	14 13	77 79	647 168
Validated screening tool to screen patients for risk of addiction	2010 2018	12 9	12 12	74 76	652 168
Validated scale to assess pain intensity	2010 2018	12 16	12 7	74 73	649 168
(NEW) Accessibility of other pharmacological agents (transdermal or sublingual buprenorphine)	2010 2018	NA 10	NA 13	NA 72	NA 168
Readily available help, such as physician mentor or 1-800 help line	2010 2018	18 17	16 11	61 68	643 167

Per cent of respondents rating usefulness of factor as 1 or 2 (not useful), 3 (neutral), or 4 or 5 (useful). ^†^ Difference is 10% or larger. Percentages may not total 100% because some respondents indicated “no opinion”. (NEW) indicate new factors added to survey in comparison to 2010. NA: not applicable.

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
