# Peer review of "Self-Reported Practices in Opioid Management of Chronic Noncancer Pain: An Updated Survey of Canadian Family Physicians"

_jcm, 2020, doi:10.3390/jcm9103304_

Round 1

Reviewer 1 Report

It would be helpful if you could include more details about the 2010 survey. Was the 2010 survey conducted via web? How many physicians participated in the 2010 survey? Was the 2010 survey a convenience sample as well?

Author Response

Comments and Suggestions for Authors

It would be helpful if you could include more details about the 2010 survey.

RESPONSE: We added the respective responses to the 2010 survey in all Tables.

Was the 2010 survey conducted via web?

RESPONSE: The 2010 survey was conducted online using Opinio, an online survey program hosted at Dalhousie University.

How many physicians participated in the 2010 survey?

RESPONSE: 710 physicians responded the 2010 survey.

Was the 2010 survey a convenience sample as well?

RESPONSE: In 2018 we used the same methods as were used in 2010 to recruit physicians. However, in 2010, the regulatory colleges were much more proactive and enthusiastic about the survey than in 2018. In 2010 the colleges sent emails to their members and added embedded links to the survey to their family physician constituents. They did not do the same in 2018.

Reviewer 2 Report

The main purpose of this study was to determine current physicians’ practices and knowledge in prescribing opioids for CNCP in relation to the updated Canadian opioid guideline, and to assess adherence to the guideline.

The rationale, methodology and implications of the main findings of the study are described in detail. The manuscript is also well written and easy to read. Nonetheless, there are some several critical weaknesses outlined in the comments to the author which make it difficult to draw meaningful conclusions

1. The manuscript would benefit from reducing some sections of the text that provide lengthy explanations of the results.

2. It would be also necessary to change the name of the “experimental section”, which can lead to confusion, into “materials & Methods”

3. In the “knowledge of opioids” section, it could increase the readability of the manuscript by specifying which questions were the most correctly selected, instead of referring to them by letters. Carefully review table 2, as well, since the statements are not associated with the letters.

4. The authors compare the new results obtained with the results of the 2010 survey in the conclusion section. I would suggest the authors to include the 2010 survey data in the figure and tables, along with the new results. This change could make more visually and intuitively the data comparison that otherwise the readers would have to read in detail.

5. One of the biggest limitations is the sample size and the lack of representation of the entire FP population. If the authors are planning to repeart the survey regularly, it could be interesting to use a respondent driven sampling in order to avoid under or over-representation of some particular FP.

Author Response

The main purpose of this study was to determine current physicians’ practices and knowledge in prescribing opioids for CNCP in relation to the updated Canadian opioid guideline, and to assess adherence to the guideline.

The rationale, methodology and implications of the main findings of the study are described in detail. The manuscript is also well written and easy to read. Nonetheless, there are some several critical weaknesses outlined in the comments to the author which make it difficult to draw meaningful conclusions

  1. The manuscript would benefit from reducing some sections of the text that provide lengthy explanations of the results.

RESPONSE: We deleted some sections of the results. Please see the document with “tracked changes”.

  1. It would be also necessary to change the name of the “experimental section”, which can lead to confusion, into “materials & Methods”

RESPONSE: We had used the Template provided in the JCM website, and it contained the word “Experimental section”. We deleted the term “Experimental Section” and replaced by “Methods and Materials”.

  1. In the “knowledge of opioids” section, it could increase the readability of the manuscript by specifying which questions were the most correctly selected, instead of referring to them by letters. Carefully review table 2, as well, since the statements are not associated with the letters.

RESPONSE: We made revisions to all TABLES and removed the letters.

  1. The authors compare the new results obtained with the results of the 2010 survey in the conclusion section. I would suggest the authors to include the 2010 survey data in the figure and tables, along with the new results. This change could make more visually and intuitively the data comparison that otherwise the readers would have to read in detail.

RESPONSE: We added all results of the 2010 survey to all TABLES.

  1. One of the biggest limitations is the sample size and the lack of representation of the entire FP population. If the authors are planning to repeart the survey regularly, it could be interesting to use a respondent driven sampling in order to avoid under or over-representation of some particular FP.

RESPONSE: Thank you for this suggestion.Bottom of Form

Reviewer 3 Report

Thank you for the opportunity to review this manuscript.

I certainly recognize the time and effort the authors have devoted to conducting their survey and preparing the manuscript.

This study reports the results of a survey of self-reported attitudes and practices in a small sample of Canadian family physicians performed in 2018/2019 and compares them with results of a similar survey performed in 2010.

While the introduction provides a summary of studies relating to the prevalence of chronic pain in Canada, prescribed opioids, and opioid-related deaths.

The summary does not distinguish between deaths due to physician- prescribed opioids, and deaths due to overdoses of illicit opioids.

The introduction does not however relate the summary to any gap in knowledge, the study is intended to address, ie what problem does this study solve.

The objectives are clearly stated, but given the comparisons with the 2010 study, as outlined in Table 1, and the discussion, it appears the authors have not included such comparisons as one of the objectives of the study.

It is therefore not clear whether this study is intended as a comparative study or not.

It appears MDPI journals use the IMRAD (Introduction, Materials and Methods, Results, and Discussions) format for the manuscript.

This Experimental Section should be labeled accordingly.

The authors state that in 2018, there were approximately 122 family medicine physicians per 100,000 population in Canada.

This statement is somewhat meaningless to the reader, unless they know the population of Canada and can calculate the number of family physicians accordingly.

I do not consider this statement should be included in the methods section, but should certainly be discussed in detail in the limitations section of the discussion section.

The authors describe their sample method as obtaining a non-probability convenience sample. As there is no way of knowing what biases are introduced by this form of sampling, authoritative epidemiological texts state it is improper to generalise the results of such samples.

The authors have described their rationale for adding new questions relating to recommendations regarding opioid doses and opioid tapering, but not to accessibility to other pharmacological agents (buprenorphine), medical cannabis use or opioid replacement therapy, given these are not recommended by the 2017 Guidelines.

The demographical variables collected are not listed in the methods section, and terms such as advanced training in pain medicine, and weak or strong opioids are not defined.

I realize weak and strong opioids are defined in the 2010 survey, but given their relevance to this study, they should be explicitly stated in the methods section. Were weak and strong opioids defined in the 2018 survey questionnaire?

Pain specialists and addiction specialists are not defined. Pain medicine is a RCPSC subspecialty, and addiction medicine is not a RCPSC specialty or subspecialty in Canada. Pain medicine did not become a subspecialty in Canada until 2012, and apart from Founding Fellows, the first RCPSC pain medicine fellowships were not awarded to 2016. There were no RCPSC pain medicine specialists at the time of the 2010 survey.

In the absence of a clear definition of pain and addiction, the validity of comparisons between the 2010 and 2018 surveys can be challenged.

It is noted that the 2017 Guidelines do not refer to pain medicine or addiction specialists, but rather Recommendation 10 provides a strong recommendation for referral to a formal multidisciplinary opioid reduction program for those experiencing serious challenges in tapering.

It might have been better to ask about access to multidisciplinary programs.

The description of frequencies surveyed for performing each practice is confusing as the categories never and <25%, and >50% and >75%, are not mutually exclusive.

It appears the surveyed categories should have been Never, 1-25%, 25-50%, 51-75% and >75%, which are then collapsed into 3 categories <25%, 25-50% and >50%, or <25%, 25-75% and >75%.

Each category on the Likert scale should be defined, and the limitations of using this ordered categorical scale discussed.

Given the number of comparisons drawn between the 2010 and current surveys, they should be compared statistically, and the methods for so doing included in the methods section.

The results are descriptive only, with comparisons with the 2010 survey included in the discussion section.

If the authors are going to compare the results of the 2010 and current surveys, then I consider the comparisons with the 2010 survey should be included within the results section, and discussed in the discussion section.

Whether or not such comparisons are valid, given the differences in the respondent demographics, need to be considered and the limitations of using historical controls discussed in the limitation section of the discussion.

If the authors are going to compare the results of the 2010 and current surveys, they should do so for all data presented in Tables.

Given the small numbers (<100) in the groups that do not prescribe any opioids and do not prescribe strong opioids, raw numbers should be provided in Table 3 and Table 4, as the percentages cited (particularly without confidence intervals), appear to inflate the numbers they represent.

The adherence to guideline Figures might better be presented in tables with raw data (and with frequencies/confidence intervals in parentheses). The responses could still be ordered as they are in the Figures.

A significant component of the discussion relates to a comparison of the results of the 2010 and the current survey.

As stated above, if this is the intent of the study, then the methods for comparing the results should be stated and the results reported in the results section.

The authors could then discuss the reasons for change, or lack of change, in the discussion section, and if change should occur, offer suggestions about how this should occur.

The limitation section of the discussion needs to be expanded to address the concerns raised by authors and reviewers.

The fundamental limitations are the small sample size, and the non-probability convenience sample. These together make it impossible to make claims about the generalizability of the results, or the validity of comparisons between the 2010 and current samples/results.

The authors of the 2010 survey justified performing that survey in part by commenting that there was little data on opioid prescribing practices of Canadian family physicians, being limited to a total of 219 respondents in three studies.

It seems the current survey would be subjected to the same criticism by the authors of the 2010 survey, some who are authors of the current survey.

While the authors of the current survey state there are similarities and differences between the demographics of the respondents of the 2010 and current surveys, they have not discussed whether or not differences are sufficient to invalidate any of the comparisons they have made.

The authors state the number of responses was only a small proportion of the 45000 family physicians (FPs) who practice in Canada. The proportion is approximately 0.59%, and given there were only 8 respondents from Ontario (14962 FPs), no respondents from Quebec (10909 FPs), and 26 respondents from British Columbia (6366 FPs), the three most populous provinces, both with respect to people and FPs, it is highly unlikely the responses from these provinces are representative.

Given there were less than 10 respondents from each of Quebec, Ontario, Manitoba, Nova Scotia, New Brunswick, Prince Edward Island and each of the Territories, the description of the survey as Canadian can be challenged.

The authors have not commented on the frequency of “no opinion” in Table 2, and how this might influence the interpretation of their results.

The authors conclude by suggesting further surveys should be conducted at 2 to 5 year intervals to monitor change, does not offer any insight in the nature or efficacy of interventions that should be used to cause change.

The survey was completed by 710 physicians in 2010, but only 256 physicians completed the current survey.

The authors have not commented on the challenges faced in trying to recruit a representative sample of family physicians, and how these challenges may be addressed if future surveys are to provide valid data to inform policies relating to opioid management of chronic non-cancer pain.

Author Response

Thank you for the opportunity to review this manuscript.

I certainly recognize the time and effort the authors have devoted to conducting their survey and preparing the manuscript.

This study reports the results of a survey of self-reported attitudes and practices in a small sample of Canadian family physicians performed in 2018/2019 and compares them with results of a similar survey performed in 2010.

While the introduction provides a summary of studies relating to the prevalence of chronic pain in Canada, prescribed opioids, and opioid-related deaths.

The summary does not distinguish between deaths due to physician- prescribed opioids, and deaths due to overdoses of illicit opioids.

RESPONSE: We added a sentence in the Introduction explaining that the majority of deaths in Canada are from nonprescribed fentanyl. Among people who died from illicit drug overdose in BC in 2015– 2017, 85.5% had opioids relevant to death on toxicology; of these, both prescribed-only opioids (2.4%) and a combination of prescribed and nonprescribed opioids (7.8%) were relatively rare. (Crabtree 2020)

The introduction does not however relate the summary to any gap in knowledge, the study is intended to address, ie what problem does this study solve.

RESPONSE: We added a sentence to the introduction.

The objectives are clearly stated, but given the comparisons with the 2010 study, as outlined in Table 1, and the discussion, it appears the authors have not included such comparisons as one of the objectives of the study.

RESPONSE: We added this as an objective of the current survey: “2) to identify changes from the survey conducted in 2010”.

It is therefore not clear whether this study is intended as a comparative study or not.

RESPONSE: Initially, our intention was only to report the current state and to identify current barriers and facilitators. However, our team agrees with the reviewers that it is a good idea to compare the current survey with the 2010 results.

It appears MDPI journals use the IMRAD (Introduction, Materials and Methods, Results, and Discussions) format for the manuscript.

RESPONSE: We had used the Template provided in the JCM website, and it contained the word “Experimental section”. We deleted the term “Experimental Section” and replaced by “Methods and Materials”.

This Experimental Section should be labeled accordingly.

RESPONSE: We changed “Experimental”  by “Methods and Meterials”.

The authors state that in 2018, there were approximately 122 family medicine physicians per 100,000 population in Canada. This statement is somewhat meaningless to the reader, unless they know the population of Canada and can calculate the number of family physicians accordingly.

I do not consider this statement should be included in the methods section, but should certainly be discussed in detail in the limitations section of the discussion section.

RESPONSE: We changed the sentence in the Introduction to:

“In 2018, there were approximately 45,000 family medicine physicians in Canada”

The authors describe their sample method as obtaining a non-probability convenience sample. As there is no way of knowing what biases are introduced by this form of sampling, authoritative epidemiological texts state it is improper to generalise the results of such samples.

RESPONSE: We acknowledge that it is improper to generalize from non-probability convenience sample. However, the fact that there were many similarities in the demographics and responses of both surveys give us more assurance that the responses are reflecting what Canadian physicians are thinking.

The authors have described their rationale for adding new questions relating to recommendations regarding opioid doses and opioid tapering, but not to accessibility to other pharmacological agents (buprenorphine), medical cannabis use or opioid replacement therapy, given these are not recommended by the 2017 Guidelines.

RESPONSE: We added information in the Methods section about the rationale for including these topics in the 2018 survey:

“We had noticed that in the 2010 survey 65% selected the wrong answer most likely because this was a tricky question, as the opioid guideline recommended to look for 30% reduction in pain to be considered clinically significant; so we revised the knowledge question “A 20% reduction in pain intensity is considered clinically significant” in the 2010 survey to “A 30% reduction in pain intensity is considered clinically significant” in the 2018 survey. There were two new questions added to the knowledge section, one about opioid replacement therapy and one about medical cannabis. None of these topics are included in the 2017 opioid guideline, and they were included as the members of our team who work with regulatory authorities have expressed interest in developing educational materials for physicians in these areas.” 

The demographical variables collected are not listed in the methods section, and terms such as advanced training in pain medicine, and weak or strong opioids are not defined.

I realize weak and strong opioids are defined in the 2010 survey, but given their relevance to this study, they should be explicitly stated in the methods section. Were weak and strong opioids defined in the 2018 survey questionnaire?

RESPONSE: Yes, in both 2010 and 2018 surveys, the question had the following explanation:

Do you prescribe weak or strong opioids for patients with Chronic Non-Cancer Pain (CNCP)?

Weak opioids- Codeine, Tramadol, Propoxyphene, Meperidine, Pentazocine

Strong opioids -Morphine, Oxycodone, Hydromorphone, Fentanyl patch, Methadone

We added this sentence in the Methods section:

The examples of weak and strong opioids were provided to them as follows: weak opioids-codeine, tramadol, propoxyphene, meperidine and pentazocine; strong opioids- morphine, oxycodone, hydromorphone, fentanyl patch and methadone. 

Pain specialists and addiction specialists are not defined. Pain medicine is a RCPSC subspecialty, and addiction medicine is not a RCPSC specialty or subspecialty in Canada. Pain medicine did not become a subspecialty in Canada until 2012, and apart from Founding Fellows, the first RCPSC pain medicine fellowships were not awarded to 2016. There were no RCPSC pain medicine specialists at the time of the 2010 survey.

In the absence of a clear definition of pain and addiction, the validity of comparisons between the 2010 and 2018 surveys can be challenged.

RESPONSE: We agree with this reviewer’s comments. The intention of the question was to gauge the wait time for patients to have access to these two types of medical expertise, which are so relevant to the field of prescribing opioids for chronic pain.

We changed the sentence in the DISCUSSION – FIRST PARAGRAPH:   

“The respondents have difficulty accessing services with expertise in pain and addiction medicine”.

We understand this reviewers’ concerns and we acknowledge that in the limitations of this manuscript. We added the following sentence in DISCUSSION - LIMITATIONS:

“In the absence of a clear definition of what is a pain specialist or an addiction specialist, it is difficult to validate the answers about wait times to specialists.”

It is noted that the 2017 Guidelines do not refer to pain medicine or addiction specialists, but rather Recommendation 10 provides a strong recommendation for referral to a formal multidisciplinary opioid reduction program for those experiencing serious challenges in tapering.

RESPONSE: We agree with this reviewer that the 2017 guideline does not provide definitions of specialties and what constitutes a formal multidisciplinary opioid reduction program.

It might have been better to ask about access to multidisciplinary programs.

RESPONSE: Thank you for the suggestion. We will take this point to the next version of this survey.

The description of frequencies surveyed for performing each practice is confusing as the categories never and <25%, and >50% and >75%, are not mutually exclusive.

It appears the surveyed categories should have been Never, 1-25%, 25-50%, 51-75% and >75%, which are then collapsed into 3 categories <25%, 25-50% and >50%, or <25%, 25-75% and >75%.

RESPONSE: There was a typo in the previous version of this manuscript. Below is the corrected text regarding the three categories:

“For these questions, the percentage of respondents performing these practices are reported in three categories: never and <25% of patients, 25% to 75% of patients, and >75% and always.”

Each category on the Likert scale should be defined, and the limitations of using this ordered categorical scale discussed.

RESPONSE:  The survey was presented in Survey Monkey as follows:

1 NOT VERY IMPORTANT

2

3

4

5 VERY IMPORTANT

We did not change anything in the manuscript as there was no definition of what is “not very important” or “very important” provided to the participants.

Given the number of comparisons drawn between the 2010 and current surveys, they should be compared statistically, and the methods for so doing included in the methods section.

RESPONSE: Thank you for this suggestion. We added all responses to the 2010 survey to this manuscript and we provided comparisons in the Methods, Results, Discussions and Tables.  Unfortunately we cannot compare using statistics because we do not have the raw data from the 2010 survey. The 2010 survey was Dr. Mike Allen’s Masters’ Thesis at Dalhousie University. The 2010 survey was approved by Dalhousie research ethics board and the raw data had been destroyed in 2017. We only have access to the results as they are presented in the publication by Allen et al in 2013.

The results are descriptive only, with comparisons with the 2010 survey included in the discussion section. If the authors are going to compare the results of the 2010 and current surveys, then I consider the comparisons with the 2010 survey should be included within the results section, and discussed in the discussion section.

RESPONSE: Thank you for this suggestion. We added the results of the 2010 survey in the METHODS, RESULTS, DISCUSSION and TABLES.

Whether or not such comparisons are valid, given the differences in the respondent demographics, need to be considered and the limitations of using historical controls discussed in the limitation section of the discussion.

RESPONSE: It is interesting to note that the demographics of the physicians who participated in the two surveys are very similar:

  • Sex distribution: In 2010 55% male. In 2018 59% male
  • Advanced training in pain management: 16% in 2010; 15% in 2018
  • Years in Practice: Almost exactly same distribution in 2010 and 2018. (See table 1)
  • Population of practice community. Very similar distribution, only difference was that 15% responded >500,000 in 2010 and 26% responded >500,000 in 2018. (See table 1)
  • Patients seen per month: Almost exactly same distribution in 2010 and 2018
  • Prescriptions for weak opioid written per month: Almost exactly same distribution in 2010 and 2018
  • Prescriptions for strong opioids written per month: Almost exactly same distribution.
  • Confidence prescribing opioids for chronic noncancer pain: Almost exactly same distribution.
  • Wait time for nonurgent referrals to pain specialists and to addiction specialists: Almost exactly same distribution, with the exception that one new alternative was added in 2018 “I don’t have this available”.

If the authors are going to compare the results of the 2010 and current surveys, they should do so for all data presented in Tables.

RESPONSE:  We added all data to the TABLEs and a comparison to the 2010 survey was conducted on an individual basis considering an absolute percentual change of 10% or larger to be a relevant change.

Given the small numbers (<100) in the groups that do not prescribe any opioids and do not prescribe strong opioids, raw numbers should be provided in Table 3 and Table 4, as the percentages cited (particularly without confidence intervals), appear to inflate the numbers they represent.

RESPONSE:  Table 4 has been eliminated (as suggested by the editor) and moved to an appendix. Table 3 contains percentages in the first three columns, and the last column is the absolute number of respondents.

The adherence to guideline Figures might better be presented in tables with raw data (and with frequencies/confidence intervals in parentheses). The responses could still be ordered as they are in the Figures.

RESPONSE: Thank you for this suggestion. We changed the two figures to Tables.

A significant component of the discussion relates to a comparison of the results of the 2010 and the current survey.

As stated above, if this is the intent of the study, then the methods for comparing the results should be stated and the results reported in the results section.

The authors could then discuss the reasons for change, or lack of change, in the discussion section, and if change should occur, offer suggestions about how this should occur.

RESPONSE: Thank you for this suggestion, we added some DISCUSSION about the reasons for change or lack of change from 2010.

The limitation section of the discussion needs to be expanded to address the concerns raised by authors and reviewers.

RESPONSE: We added more topics to the DISCUSSION – LIMITATIONS section.

The fundamental limitations are the small sample size, and the non-probability convenience sample.

RESPONSE: These limitations are included in the DISCUSSION.

These together make it impossible to make claims about the generalizability of the results, or the validity of comparisons between the 2010 and current samples/results.

RESPONSE: This is explained in the ABSTRACT and DISCUSSION.

The authors of the 2010 survey justified performing that survey in part by commenting that there was little data on opioid prescribing practices of Canadian family physicians, being limited to a total of 219 respondents in three studies.

It seems the current survey would be subjected to the same criticism by the authors of the 2010 survey, some who are authors of the current survey.

RESPONSE: We agree, our sample size was small.

While the authors of the current survey state there are similarities and differences between the demographics of the respondents of the 2010 and current surveys, they have not discussed whether or not differences are sufficient to invalidate any of the comparisons they have made.

RESPONSE: We added this information to the DISCUSSION section – FIRST PARAGRAPH.

The authors state the number of responses was only a small proportion of the 45000 family physicians (FPs) who practice in Canada. The proportion is approximately 0.59%, and given there were only 8 respondents from Ontario (14962 FPs), no respondents from Quebec (10909 FPs), and 26 respondents from British Columbia (6366 FPs), the three most populous provinces, both with respect to people and FPs, it is highly unlikely the responses from these provinces are representative.

RESPONSE: We agree with this reviewer. We had originally thought that we would obtain enough numbers from each Province to do some analyses between Provinces. However, this was clearly not possible.

Given there were less than 10 respondents from each of Quebec, Ontario, Manitoba, Nova Scotia, New Brunswick, Prince Edward Island and each of the Territories, the description of the survey as Canadian can be challenged.

RESPONSE: We could have excluded all respondents from the Provinces with small number of responses, but we thought these people took time from their busy practices to answer our survey (approximately 30 minutes) and they were not compensated for their time. Our group was confident that including other Provinces would not introduce any bias, given that the 2010 and 2017 opioid guidelines were distributed and implemented in all Canadian Provinces.

The authors have not commented on the frequency of “no opinion” in Table 2, and how this might influence the interpretation of their results.

RESPONSE: The team that selected the questions for the 2010 review was slightly different from the team in 2018, and we had thought about eliminating the option “No Opinion” from the 2018 survey. However, making too many changes would eliminate our ability to compare the two surveys, therefore, we opted to leave the alternative “no opinion” in 2018. We added a sentence in RESULTS and DISCUSSION.

The authors conclude by suggesting further surveys should be conducted at 2 to 5 year intervals to monitor change, does not offer any insight in the nature or efficacy of interventions that should be used to cause change.

RESPONSE: We added a sentence in the DISCUSSION

This national survey is an attempt to assess if the release of opioid guidelines have been achieving the changes that they proposed to make.

The survey was completed by 710 physicians in 2010, but only 256 physicians completed the current survey.

The authors have not commented on the challenges faced in trying to recruit a representative sample of family physicians, and how these challenges may be addressed if future surveys are to provide valid data to inform policies relating to opioid management of chronic non-cancer pain.

RESPONSE: We added a sentence to the DISCUSSION:

It is important to seek support and endorsement of organizations such as the College of Family Physicians of Canada and the medical regulatory authorities.

Round 2

Reviewer 2 Report

The authors have addressed most of the issuses rised

The quality of the manuscript has been improved with the changes included, although I still recommend the authors to consult an statistician in order to improve the data collection and analysis in future researches

I would recommend the authors to change the sentence "It would be informative to repeat this survey at regular intervals in two to five years to detect changes over time; however this will have to take account of the changes in format and recommendations of updated guidelines" (line 449), and include some statement regarding how to improve sampling method

Author Response

Reviewer #2

The authors have addressed most of the issuses rised

  • The quality of the manuscript has been improved with the changes included, although I still recommend the authors to consult an statistician in order to improve the data collection and analysis in future researches

RESPONSE: Thank you for this recommendation. We will seek funding to conduct this survey in the future, this will enable us to develop advertising materials to reach a broader sample of family physicians in Canada. With funding, we would also be able to buy mailing lists of physicians and send email directly to their inboxes. With funding, we will be able to reach a proper random sample instead of a convenience sample. Unfortunately, we did not have external funds to conduct the survey in 2018. The survey in 2010 was administered by all the Colleges and Medical Regulatory Authorities in Canada and we had a dedicated Masters Student who was doing the survey as material for his thesis defense (Dr. Michael Allen). In the future, we will hire a project coordinator to assist our team to formalize a partnership with professional organizations of physicians that could help us to promote and advertise the survey. We learned the lesson that without a formal project coordination and funds to advertise we would not be able to reach a large random sample of family doctors.

  • I would recommend the authors to change the sentence "It would be informative to repeat this survey at regular intervals in two to five years to detect changes over time; however this will have to take account of the changes in format and recommendations of updated guidelines" (line 449), and include some statement regarding how to improve sampling method.

RESPONSE: Thank you for this suggestion. We changed to the following:

“It would be informative to repeat this survey in two to five years to obtain updated status and develop current educational materials for physicians.  This will involve formalizing partnerships with physician professional, educational, and regulatory organizations to advertise and encourage responses to the survey thereby reaching a broader sample of family physicians. It will also have to consider changes in format and recommendations of updated guidelines.”

Reviewer 3 Report

The authors have addressed most of my concerns.

They have not clearly stated the reason for repeating the study in the introduction.

They have arbitrarily chosen a 10% difference in outcome as significant, without justifying why they have made this choice or not undertaken a formal statistical comparison.

On the other hand, they may feel the current study population is significantly different from the previous study, in which case it could be argued that no comparisons should be drawn.

The authors should consider consulting a statistician regarding their analysis.

The authors have improved their explanation of differences between the two studies in the discussion section.

The authors have improved their acknowledgement of limitations.

I consider the use of the term "small proportion" when comparing their sample size with the population of Canadian GPs to be overstating the significance of their sample 265/45000 = 0.6%, and hence the generalisability of the results.

They should consider using "very small" or "tiny".

Although the authors have recognised the need for knowledge translation, exchange and education, they have not commented on how this might be achieved.

I do not consider the authors have made a compelling argument for the need to reproduce this study in the future. Otherwise, how would they suggest medical regulatory bodies or professional organizations be induced to endorse future surveys so that a representative population of physicians were surveyed.

Finally, why have the authors not recommended that future surveys be designed to obtain a nationally representative sample of physicians, rather than be subject to the same methodological limitations as the current study?

Author Response

Reviewer #3 (please see attached file with figures in this response)

The authors have addressed most of my concerns.

1- They have not clearly stated the reason for repeating the study in the introduction.

RESPONSE:  We added the following sentence in the Introduction Lines 58-62

“There are various groups developing toolkits, educational resources and materials to implement the guideline in Canada, and for this reason it is important to have access to an updated survey to inform the development of these knowledge transfer and exchange materials.”

2- They have arbitrarily chosen a 10% difference in outcome as significant, without justifying why they have made this choice or not undertaken a formal statistical comparison.

On the other hand, they may feel the current study population is significantly different from the previous study, in which case it could be argued that no comparisons should be drawn.

The authors should consider consulting a statistician regarding their analysis.

RESPONSE: Thank you for pointing that out to us. We missed the justification in the previous revision of this paper.

We added two references to justify the choice of 10% in the current version of this manuscript. (see table below)

The following sentence is added to the justification in the manuscript: Line 135

“Previous surveys conducted with opioid prescribing physicians showed that differences as low as 7.3% are relevant and significant.[11, 12]”

In 2018, Andrea Furlan et al published a systematic review of strategies to improve appropriate use of opioids and to reduce opioid use disorder and deaths from prescription opioids. https://www.tandfonline.com/doi/full/10.1080/24740527.2018.1479842 This systematic review included 65 studies, of these, many assessed physicians behaviours related to prescribing opioids before and after some sort of intervention. Below is a summary of two articles included in the Furlan et al 2018 review that were used to calculate the 10% as a significant difference.

Study

Authors’ conclusions

Study findings

Lofwall MR, Wunsch MJ, Nuzzo PA, Walsh SL.

Efficacy of continuing medical education to reduce the risk of buprenorphine diversion.

J Subst Abuse Treat. 2011 Oct;41(3):321-9. doi: 10.1016/j.jsat.2011.04.008. Epub 2011 Jun 12. PMID: 21664789.

The results show that physicians had limited addictions training. Knowledge and practice behaviors significantly improved after the CME, which should enhance the quality of OBOT and may decrease risk of buprenorphine misuse and diversion from their practices. Mandatory CME targeting

Our interpretation is that changes occur in a variety of directions and magnitude. By selecting 10% we are in concordance with the authors of this paper.

Katzman JG, Comerci GD, Landen M, Loring L, Jenkusky SM, Arora S, Kalishman S, Marr L, Camarata C, Duhigg D, Dillow J, Koshkin E, Taylor DE, Geppert CM.

Rules and values: a coordinated regulatory and educational approach to the public health crises of chronic pain and addiction.

Am J Public Health. 2014 Aug;104(8):1356-62. doi: 10.2105/AJPH.2014.301881. Epub 2014 Jun 12. PMID: 24922121; PMCID: PMC4103251.

“The results demonstrate that an innovative coalition of an academic medical center, the state department

of health, the medical and

pharmacy licensing boards, the Project ECHO Institute, and the state legislature can develop a mandated CME requirement that

significantly impacts the knowledge, attitudes, and self-efficacy of practitioners with regard to best practices in pain management and opioid prescribing.”

Our interpretation is that even a small change of 7.3% could mean a significant change with a large effect size. Therefore, our team selected 10% as a conservative minimal absolute change as meaningful for educators who are developing continuing medical education (CME) materials.

Regarding the choice for not conducting statistical testing between the 2010 and 2018 surveys, our rationale includes the following:

  1. The intention of the 2018 survey was mainly to determine current physicians’ practices and knowledge in prescribing opioids for CNCP in relation to the updated Canadian opioid guideline. There are various groups developing toolkits, educational resources and materials to implement the guideline in Canada, and for this reason it is important to have access to an updated survey to inform the development of these knowledge transfer and exchange materials. In the original submission of this paper we had not included the results of the 2010 survey because those results are outdated and not relevant to people developing materials to implement the guideline in Canada.
  2. We do not think it is appropriate to calculate p values for each question of the survey because we would generate a very large amount of p values, which by chance alone, some would be significant, even if we use a lower threshold of significance.
  3. Another reason for not doing statistical comparisons is that we do not have a hypothesis to test. All survey questions are equally important and relevant to the stakeholders.
  4. Third, the sample of the 2010 survey is 2.5 times larger than the 2018 survey and this imbalance could generate spurious significant values. The variance around the estimates in 2010 would have to be re-calculated because we did not retain the original data set. As per research ethics, we had to destroy the raw data in 2017. Calculations of variance for binary proportions would not be a big problem. However, all questions in the survey are not binary as they have multiple alternatives to select from, and in this case the recalculated variances could be biased.
  5. The two samples are very similar in terms of sex, gender, advanced training in pain management, years in practice, community size, patients seen per month, prescriptions of opioids written per month, confidence prescribing opioids for chronic noncancer pain, and access to specialist. The only major difference is the geographical distribution of physicians. However, given that the two guidelines were nationally distributed and implemented in Canada, we think that all physicians in Canada had the same exposure to the Canadian Opioid Guideline released in 2010 and updated in 2018.
  6. We did consult with the senior statistician at the Institute for Work & Health in Toronto. She corroborated with our team’s decision for not conducing multiple comparisons given the reasons explained above.

5- The authors have improved their explanation of differences between the two studies in the discussion section.

RESPONSE: Thank you.

6- The authors have improved their acknowledgement of limitations.

RESPONSE: Thank you.

7- I consider the use of the term "small proportion" when comparing their sample size with the population of Canadian GPs to be overstating the significance of their sample 265/45000 = 0.6%, and hence the generalisability of the results.

They should consider using "very small" or "tiny".

RESPONSE: We changed the sentence to “very small proportion”.  Line 390

9- Although the authors have recognised the need for knowledge translation, exchange and education, they have not commented on how this might be achieved.

RESPONSE: Thank you for these suggestions. We revised the sentence as follows: (Line 408)

This indicates there is still a great need for knowledge translation, exchange and education (KTE/E) in how to prescribe opioids for patients with chronic pain. Authors of this paper are also members of the Canadian Opioid Guideline National Faculty, a cross-Canada group of stakeholders tasked with the ongoing KTE/E in Canada.

10- I do not consider the authors have made a compelling argument for the need to reproduce this study in the future. Otherwise, how would they suggest medical regulatory bodies or professional organizations be induced to endorse future surveys so that a representative population of physicians were surveyed.

RESPONSE: The Canadian Opioid Guideline is expected to be updated in the next year or two. Once the new version of the guideline is released, the results of the 2018 survey will be irrelevant to people who are developing materials to implement the new guideline. Therefore, there will be a need to update the survey again.

11- Finally, why have the authors not recommended that future surveys be designed to obtain a nationally representative sample of physicians, rather than be subject to the same methodological limitations as the current study?

RESPONSE: We will seek funding to conduct this survey in the future, this will enable us to develop advertising materials to reach a broader sample of family physicians in Canada. With funding, we would also be able to buy mailing lists of physicians and send email directly to their inboxes. With funding, we will be able to reach a proper random sample instead of a convenience sample. Unfortunately, we did not have external funds to conduct the survey in 2018. The survey in 2010 was administered by all the Colleges and Medical Regulatory Authorities in Canada and we had a dedicated Masters Student who was doing the survey as material for his thesis defense (Dr. Michael Allen). In the future, we will hire a project coordinator to assist our team to formalize a partnership with professional organizations of physicians that could help us to promote and advertise the survey. We learned the lesson that without a formal project coordination and funds to advertise we would not be able to reach a large random sample of family doctors.

We changed the paragraph in the Discussion as follows: (Line 411)

“It would be informative to repeat this survey in two to five years to obtain updated status and develop current educational materials for physicians. This will involve formalizing partnerships with physician professional, educational, and regulatory organizations to advertise and encourage responses to the survey thereby reaching a broader sample of family physicians. It will also have to consider changes in format and recommendations of updated guidelines.”
